# In-depth characterization and comparison of the N-glycosylated proteome of two-dimensional- and three-dimensional-cultured breast cancer cells and xenografted tumors

Yonghong Mao[1,2], Yang Zhao[3], Yong Zhang[ID][4,5,6]*, Hao Yang[4,5,6]*

**1** Institute of Thoracic Oncology, West China Hospital, Sichuan University, Chengdu, China, **2** Department of Thoracic Surgery, West China Hospital, Sichuan University, Chengdu, China, **3** Mass Spectrometry Engineering Technology Research Center, Center for Advanced Measurement Science, National Institute of Metrology, Beijing, China, **4** Key Lab of Transplant Engineering and Immunology, MOH, West China Hospital, Sichuan University, Chengdu, China, **5** Institutes for Systems Genetics, West China Hospital, Sichuan University, Chengdu, China, **6** Frontiers Science Center for Disease-related Molecular Network, West China Hospital, Sichuan University, Chengdu, China

* nankai1989@foxmail.com (YZ); yanghao@scu.edu.cn (HY)

**Data Availability Statement:** All relevant data are within the manuscript and its Supporting information files.

## Abstract

Native intact N-glycopeptide analysis can provide access to the comprehensive characteristics of N-glycan occupancy, including N-glycosites, N-glycan compositions, and N-glycoproteins for complex samples. The sample pre-processing method used for the analysis of intact N-glycopeptides usually depends on the enrichment of low abundance N-glycopeptides from a tryptic peptide mixture using hydrophilic substances before LC-MS/MS detection. However, the number of identified intact N-glycopeptides remains inadequate to achieve an in-depth profile of the N-glycosylation landscape. Here, we optimized the sample preparation workflow prior to LC-MS/MS analysis by systematically comparing different analytical methods, including the use of different sources of trypsin, combinations of different proteases, and different enrichment materials. Finally, we found that the combination of Trypsin (B)/Lys-C digestion and zwitterionic HILIC (Zic-HILIC) enrichment significantly improved the mass spectrometric characterization of intact N-glycopeptides, increasing the number of identified intact N-glycopeptides and offering better analytical reproducibility. Furthermore, the optimized workflow was applied to the analysis of intact N-glycopeptides in two-dimensional (2D) and three-dimensional (3D)-cultured breast cancer cells *in vitro* and xenografted tumors in mice. These results indicated that the same breast cancer cells, when cultured in different microenvironments, can show different N-glycosylation patterns. This study also provides an interesting comparison of the N-glycoproteome of breast cancer cells cultured in different growth conditions, indicating the important role of N-glycosylated proteins in cancer cell growth and the choice of the cell culture model for studies in tumor biology and drug evaluation.

**Funding:** (Yong Zhang) National Natural Science Foundation of China (grant number 31901038) and the China Postdoctoral Science Foundation (2019M653438).

**Competing interests:** The authors have declared that no competing interests exist.

## Introduction

N-glycosylation is the most common post-translational modification of proteins in human cell lines. Adhesive N-glycans play significant roles in various biological processes such as immune defense, cell growth and communication [1]. The glycoproteins that exist on the transmembrane domain or the cell surface are easily accessible to therapeutic drugs, vaccines, and antibodies [2]. Hence, N-glycosylation analysis has become a vital tool in understanding the factors that affect potential treatment benefits, and therefore needs an optimized workflow for extensive investigation.

In the past years, a variety of commercially available proteases, combinations of different enzymes, and various enrichment materials have been developed [3]. However, because of the microheterogeneity and macroheterogeneity of N-glycosylation, none of these methods or materials have so far been able to identify all intact N-glycopeptides in complex biological samples [4]. Currently, there is a lack of systematic intact N-glycopeptide analyses and comparisons using these newly developed reagents or materials. Fortunately, some reports have shown that the combined enzyme digestion approach (such as Lys-C/Arg-C or Lys-C/trypsin) could be a better choice in both qualitative and quantitative proteomics studies [5, 6]. Other reports have shown that two-step protease digestion (Lys-C-trypsin) could be helpful for accurate glycosite identification and improvement in glycoprotein sequence coverage [7, 8]. Besides, some novel materials with multiple functions have been developed and are already available commercially [9–11]. Hence, these methods and materials provide the conditions for us to develop an optimized workflow for in-depth N-glycosylation analysis.

Cell culture models and xenografted tumors in mice are valuable resources for researchers to investigate biological mechanisms in humans [12]. The most common cell culture method for propagating cells is in 2D because of its simple operation and low cost. Alternatively, 3D cell cultures mimic the *in vivo* chemical microenvironments more closely and are widely used to evaluate the effect of new drug compounds [13, 14]. Based on this, quantitative proteomic and phosphoproteomic comparisons of 2D and 3D cancer cell culture models have been reported. It was revealed that the differences in protein expression and phosphorylation between 2D- and 3D-cultured cells, and 3D cultures cells are better representations of *in vivo* conditions [15, 16]. However, it is worth noting that the target proteins of drug action are often glycosylated [17]. The cellular environment can affect the N-glycosylation expression patterns and consequently, drug evaluation results [18]. It is therefore necessary to compare the N-glycosylation patterns of proteins from 2D and 3D cancer cell culture models and xenografted tumors in mice.

To achieve this goal, we optimized the experimental conditions for the large-scale identification of native intact N-glycopeptides using a combination of Trypsin (B)/Lys-C for digestion and Zic-HILIC for enrichment. Then, the optimized method was further applied to explore the N-glycosylation patterns from 2D and 3D-cultured breast cancer cells *in vitro* and xenografted tumors in mice. Using high-resolution mass spectrometry and professional software to analyze intact N-glycopeptides, we revealed that the same breast cancer cells cultured in different microenvironments showed different N-glycosylation patterns. The raw mass spectrometry data obtained in this study are publicly accessible at ProteomeXchange (ProteomeXchange. com) under Accession Number PXD020254.

## Materials and methods

### Chemicals

Venusil HILIC sorbent and C18 reverse-phase medium were purchased from Agela Technologies (Tianjin, China). Zwitterionic HILIC (Zic-HILIC) was purchased from Fresh Bioscience

(Shanghai, China). The C8 extraction disks were purchased from 3M Empore™ (St. Paul, MN, USA). Sequencing grade trypsin (A) was obtained from Promega (Madison, WI, USA). Sequencing grade trypsin (B) and Lys-C were obtained from Enzyme & Spectrum (Beijing, China). Dithiothreitol (DTT), iodoacetamide (IAA), formic acid (FA), trifluoroacetic acid (TFA), Tris base and urea were purchased from Sigma (St. Louis, MO, USA). Acetonitrile (ACN) and acetic acid (HAc) were purchased from Merck (Darmstadt, Germany). The quantitative Bradford protein assay kit and colorimetric peptide assay kit were purchased from Thermo Fisher Scientific (Waltham, MA, USA). All other chemicals and reagents of the best available grade were purchased from Sigma-Aldrich (St. Louis, MO, USA) or Thermo Fisher Scientific (Waltham, MA, USA).

## Cell culture and xenografted tumor model

Human hepatoma cells (HepG2) and breast cancer cells (MDA-MB-231) were obtained from the American Type Culture Collection (ATCC). HepG2 and MDA-MB-231 cells were identified through Short Tandem Repeat (STR) profiling. Cells were cultured in Dulbecco's modified Eagle's medium (DMEM) with 10% fetal bovine serum and grown in 5% $CO_2$ at 37˚C. For the 2D monolayer cultures, $2 \times 10^5$ cells were seeded in a Petri dish. For the 3D spheroid cultures, $3 \times 10^3$ cells were seeded in 96-well ultra low attachment plates with Corning Matrigel matrix to form multicellular aggregates.

All animal experiments were performed with a protocol approved by the Animal Care and Use Committees at the West China Hospital, Sichuan University in compliance with their guidelines and animal welfare standards. Four to five-week-old female BALB/c nude mice were purchased from Beijing Charles River Laboratories. Mice were housed in the specific-pathogen-free (SPF) animal facility at room temperature (22˚C), containing standard plastic cages on ventilated racks providing filtered air, 12 h light-dark cycle, and 30–70% humidity. Commercially-available laboratory rodent diets were provided ad libitum. Cages and diets were changed weekly. Mice were accustomed to the housing conditions for one week before tumor cell inoculation. $2 \times 10^6$ cells were subcutaneously injected into the left flank region of mice (n = 5). Mouse activity, diet intake, and body weight were observed and recorded every day for assessing mouse health. The tumor size was recorded every other day and the tumor volume was calculated as length×width$^2$×0.5. When tumor grafts reached approximately 200 mm$^3$, the mice (n = 2) were euthanized with isoflurane and tumors were collected for further use.

## Protein extraction

At 80% confluence, 2D and 3D cell cultures were harvested and aspirated from the medium with ice-cold phosphate buffer saline (PBS). The cells were resuspended in ice-cold lysis buffer (8 M urea, 50 mM Tris/HCl, pH 8.5, 1% protease inhibitor cocktail) and sonicated for 5 min using an ultrasonic processor (Kunshan, China). The lysate was centrifuged at $13,000 \times g$ for 20 min at 4˚C. The xenografted tumors in mice were put in a 2-mL lysing matrix tube and soaked in ice-cold lysis buffer (8 M urea, 50 mM Tris/HCl, pH 8.5, 1% protease inhibitor cocktail), homogenized using a tissue lyser (Hangzhou, China. speed = 6.5 m/s; time = 60 s; number of cycles = 3), and sonicated for 5 min using an ultrasonic processor. The homogenate was centrifuged at $13,000 \times g$ for 15 min at 4˚C. The supernatant was then collected and stored as a soluble fraction at −80˚C until use. Total protein concentration was determined by using a Bradford protein assay kit. Bovine serum albumin was used as a protein standard.

## Reduction, alkylation and digestion

The extracted proteins were proteolyzed using the filter-aided sample preparation (FASP) protocol. Briefly, 200 μg of protein was diluted with a UA solution (8 M urea in 0.1 M Tris-HCl, pH 8.5) and added to a 30-kDa filter tube. After centrifuging at 13,000 × *g* for 15 min at 25˚C, 200 μL of UA solution with 20 mM DTT was added, and the reduction reaction was carried out for 4 h at 37˚C. The solution was removed by centrifugation, and 200 μL of UA solution with 50 mM iodoacetamide (IAA) was added and incubated in the dark for 1 h at 25˚C. After washing with 200 μL UA two times and 200 μL of 50 mM ammonium bicarbonate three times and centrifugation at 13,000 × *g* for 15 min at 25˚C, 5 μg of trypsin (B)/Lys-C were added to each filter tube, respectively. After incubating for 24 h at 37˚C, the filter tubes were washed twice with 100 μL of water by centrifugation at 13,000 × *g* for 15 min at 25˚C. The flow-through fractions were collected. The peptide concentration was determined using a quantitative colorimetric peptide assay kit based on absorbance at a wavelength of 480 nm. The peptide mixtures were freeze-dried and then stored at −80˚C until use.

## Intact N-glycopeptides enrichment

Intact N-glycopeptides were enriched by HILIC (Venusil, Tianjin) or Zic-HILIC (Fresh Bioscience, Shanghai). Specifically, 100 μg of tryptic peptides were resuspended in 100 μl of 80% ACN/0.2% TFA solution. Then, 5 mg of HILIC or Zic-HILIC was washed three times for 10 min each with 100 μl of 0.1% TFA and 80% ACN/0.2% TFA, followed by sample loading and rotation for 2 h at 37˚C. Lastly, the mixture was transferred to a 200 μL pipet tip that was packed with a layer of C8 membrane and washed twice with 70μl of 80% ACN/0.2% TFA. Intact N-glycopeptides bound to the HILIC or Zic-HILIC column were eluted three times with 70 μL of 0.1% TFA. The pooled eluent was dried via SpeedVac and resuspended in 20 μl of 0.1% FA for HPLC fractionation.

## LC-MS/MS analysis

All samples were analyzed with LC-MS/MS using an Orbitrap Fusion Lumos Mass Spectrometer (Thermo Fisher, USA). The parameters were set as described previously. Briefly, peptides were dissolved in 0.1% FA and separated on a 75-μm-inner-diameter column with a length of 20 cm (ReproSil-Pur C18-AQ, 1.9 μm; Dr Maisch) over a 78-min gradient (buffer A, 0.1% FA in water; buffer B, 0.1% FA in 80% ACN) at a flow rate of 300 nL/min. The MS1 was performed employing scan range (m/z) of 800–2000 at an Orbitrap resolution of 120,000. The RF lens, AGC target, maximum injection time, and exclusion duration were 30%, $2.0\ e^5$, 100 ms, and 15 s, respectively. The MS2 was performed employing an isolation window (m/z) of 2 at an Orbitrap resolution of 15,000. The AGC target, maximum injection time, and HCD collision arbitrary value of $5.0\ e^5$, 250 ms, and 30%, respectively. The stepped collision mode was turned on with an energy difference of ±10% (20-30-40%).

## Data analysis

The raw data files of intact N-glycopeptides were searched against the human Uniprot database (version 2015_03, 20,410 entries) using Byonic software (version 3.6.0, Protein Metrics, Inc.) as described previously [19]. Parameters are listed as follows: mass tolerance for precursors and fragment ions were set as ± 10 ppm and ± 20 ppm, respectively. Two missed cleavages sites were allowed for enzyme digestion. The fixed modification was Carbamidomethyl (C). Variable modifications contained Oxidation (M) and Acetyl (Protein N-term). Additionally, the "N-glycan 309 mammalian no sodium" was specified as the N-glycan modification for all searches. The protein

database options, including the decoy and common contaminants, were ticked. All other settings were set at the default values, and protein groups were filtered to 1% FDR based on the number of hits obtained for searches against these databases. Stricter quality control methods for intact N-glycopeptide identification included a score of over 200 and at least 7 amino acids to be considered an intact N-glycopeptide. The heat map drawing, Gene Ontology (GO) functional analysis, and pathway analysis were performed using MSigDB gene sets in R packages.

### Statistical analysis

The number of intact N-glycopeptides, N-glycan compositions, and N-glycoproteins identified from HepG2 cell lines were analyzed by Student's t-test for statistical comparison between two groups. Three technical replicates for each group were performed. The number of intact N-glycopeptides, N-glycan compositions, and N-glycoproteins identified from 2D- and 3D-cultured breast cancer cells in vitro and xenografted tumors in mice were analyzed by Student's t-test for statistical comparison between two groups and one-way ANOVA test among three groups. Each group contains six replicates (two biological replicates and three technical replicates for each group). Data were presented as means ± SD. *P*-value < 0.05 was considered significant. Student's t-test and ANOVA test were performed in SPSS v. 19.0 (IBM Corp., Armonk, NY, USA). Other analyses were run in R or a customized in-house platform (http://www.omicsolution.org/wu-kong-beta-linux/main/) [20].

## Results and discussion

### Optimized workflow for in-depth analysis of intact N-glycopeptides

Currently, common methods for sample preparation of tryptic N-glycopeptide enrichment using HILIC or other materials usually obtain relatively few high-confidence intact N-glycopeptide species [21, 22]. Different protease combinations have also been used in proteomic research [5]; however, this strategy was rarely applied to investigate intact N-glycopeptides. Therefore, it is necessary to optimize and integrate existing workflows for in-depth analysis of intact N-glycopeptides. As shown in Fig 1A, we systematically compared different proteases and their combinations from different sources and different enrichment materials using the common HepG2 liver cancer cell line. Specifically, the same amount of proteins extracted from HepG2 cells were denatured, reduced, and alkylated. The same amount of trypsin (A), trypsin (B), and trypsin (B)/Lys-C mixtures were added to digest proteins. Finally, intact N-glycopeptides were enriched using HILIC or Zic-HILIC and analyzed by SCE-HCD-MS/MS. The reported results were obtained from at least three independent experiments. The commercial software, Byonic, was chosen for the identification of intact N-glycopeptides containing N-X-S/T (X≠P) sequons [19]. Quality control standards included <1% FDR at the protein level, a Byonic score >200, and >6 amino acids for the high-confidence identification of intact N-glycopeptides. The spectra were checked manually.

By comparing these identification results, we found that the Trypsin (B)/Lys-C combination produced more intact N-glycopeptides than individual trypsin (A) or trypsin (B) (*P*<0.05, Student's t-test) with good reproducibility (Fig 1B). Furthermore, trypsin (B) was better than trypsin (A) in producing intact N-glycopeptides (*P*<0.05, Student's t-test). Zic-HILIC can enrich more intact N-glycopeptides than HILIC (*P*<0.05, Student's t-test). Hence, the optimized workflow and the combination of Trypsin (B)/Lys-C for digestion and Zic-HILIC for enrichment (Trypsin (B)/Lys-C-Zic-HILIC group) can realized in-depth identification (an average of 2221 intact N-glycopeptides in each repeat) with high analytical reproducibility, and had nearly twice as many N-glycopeptides identified

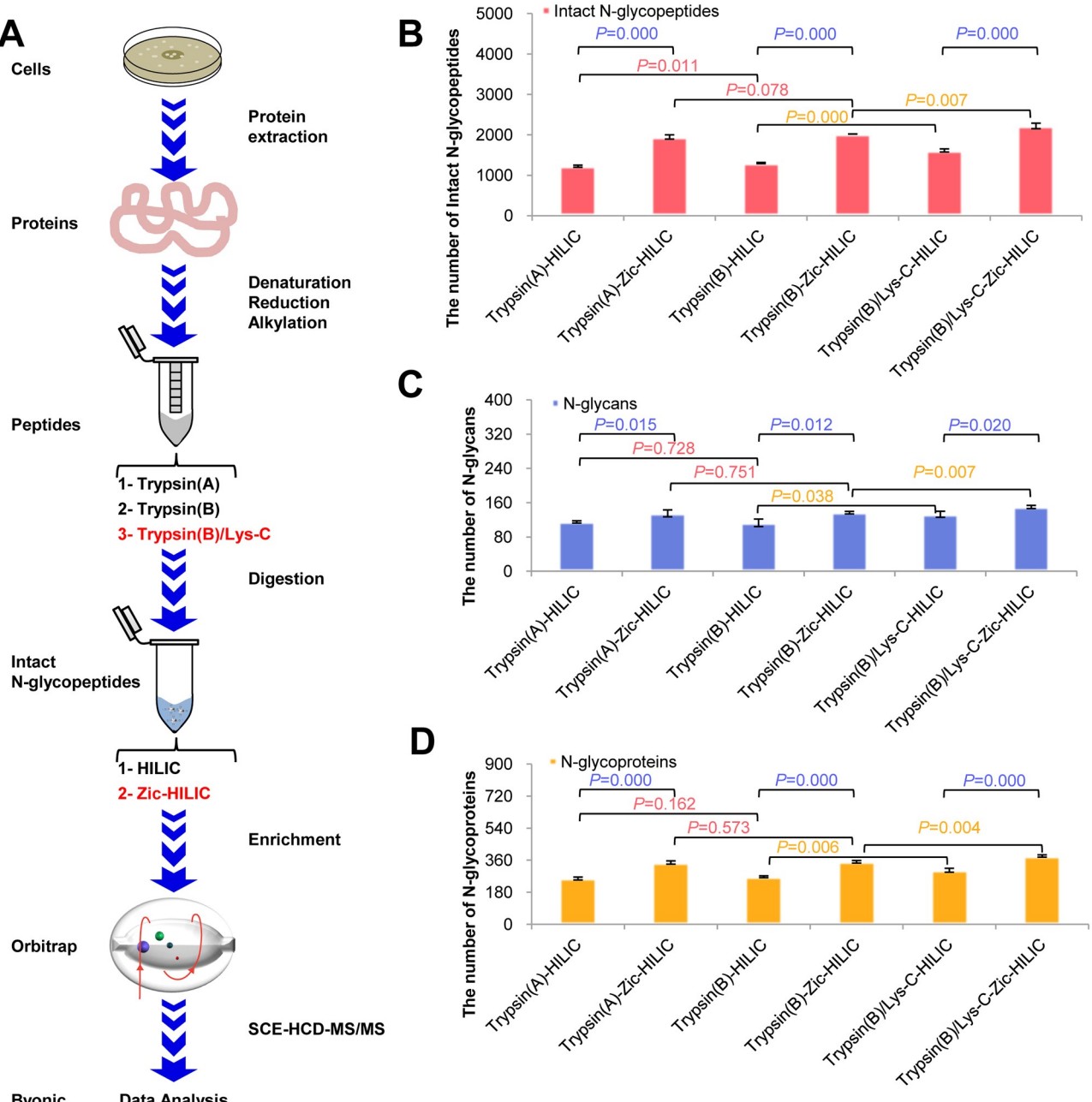

**Fig 1. Optimized workflow for in-depth intact N-glycopeptide analysis and comparative results. A.** Workflow for comparison of different experimental methods. **B.** Comparison of intact N-glycopeptides. **C.** Comparison of N-glycans. **D.** Comparison of N-glycoproteins.

compared to the Trypsin (A)-HILIC group (an average of 1223 intact N-glycopeptides in each repeat). We further compared the identification of N-glycans and N-glycoproteins using the optimized workflow. Results showed that this workflow could also identify a larger number of N-glycans (Fig 1C) and N-glycoproteins (Fig 1D). The complete list of identified intact N-glycopeptides using different methods is detailed in S1 Table.

## N-glycosylation profiling of 2D and 3D breast cancer cell cultures and cell xenografted tumors

The intact N-glycopeptides from 2D- and 3D-cultured breast cancer cells *in vitro* and xenografted tumors in mice were analyzed using the optimized workflow (Fig 2A). We identified adequate numbers of intact N-glycopeptides from these three sample groups. A total of 740 N-glycoproteins, including 7229 N-glycopeptides and 269 N-glycan compositions, were

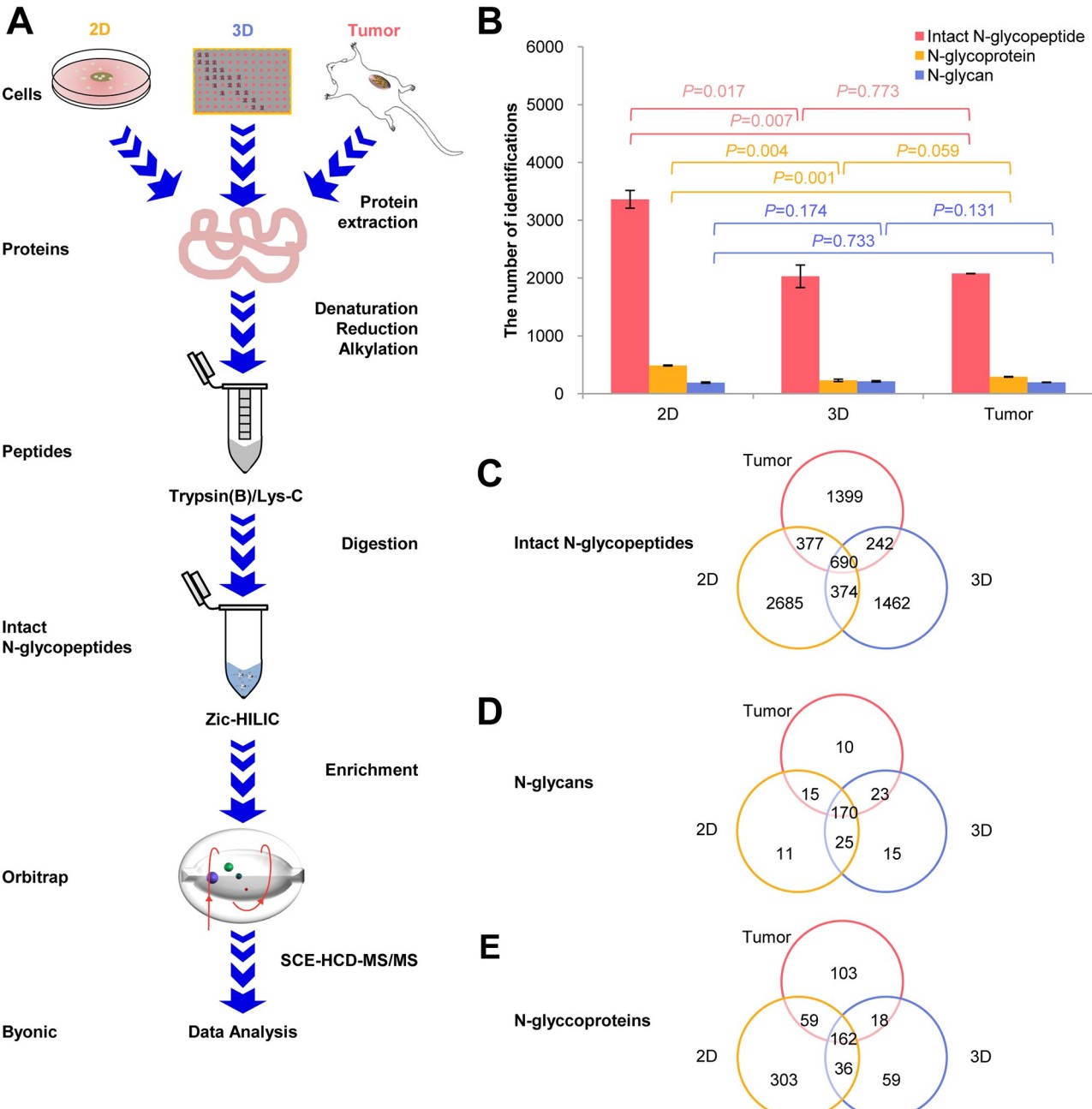

**Fig 2. N-glycosylation profiling of 2D and 3D breast cancer cell cultures and xenografted tumors using the optimized workflow. A.** Optimized workflow for comparison of the intact N-glycopeptides from the three groups. **B.** Comparison of N-glycosylation in the three groups. **C.** Venn diagram of the identified intact N-glycopeptides. **D.** Venn diagram of the identified N-glycoproteins. **E.** Venn diagram of the identified N-glycans.

identified (S2 Table). The overlap between biological replicates indicated good reproducibility of the experiment. For example, 416 unique N-glycoproteins (74.3%) with 166 unique N-glycan compositions (75.1%) and 2601 unique intact N-glycopeptides (63.0%) were identified in both 2D-cultured breast cancer cells (S1A Fig), while 187 unique N-glycoproteins (68.0%), including 189 unique N-glycan compositions (81.1%) and 1298 unique intact N-glycopeptides (46.9%), were identified in both 3D-cultured breast cancer cells (S1B Fig). Furthermore, 242 unique N-glycoproteins (70.8%), with 175 unique N-glycan compositions (80.3%) and 1149 unique intact N-glycopeptides (53.5%), were identified in both xenografted tumors (S1C Fig). There is relatively little overlap (~55.0%) in intact N-glycopeptides between two biological replicates that might be caused by the microheterogeneity of N-glycosylation. The complete list of identified intact N-glycopeptides of these samples is provided in S2 Table.

Furthermore, we compared the number of intact N-glycopeptides, N-glycans, and N-glycoproteins identified from the three groups. As shown in Fig 2B, the 2D group contained more intact N-glycopeptides and N-glycoproteins compared with the 3D group or the tumor xenograft group ($P<0.05$, Student's t-test). While the number of intact N-glycopeptides or N-glycoproteins between the 3D group and the tumor xenograft group were similar ($P>0.05$, Student's t-test), it is worth noting that the number of N-glycan compositions between the three groups were not significantly different either ($P>0.05$, Student's t-test). We believe that the most of the 309 mammalian N-glycans can be identified from these three groups using our optimized workflow. We then compared the types of intact N-glycopeptides, N-glycans, and N-glycoproteins identified from the three groups. Because of the microheterogeneity in N-glycosylation, the types of intact N-glycopeptides and N-glycan compositions ($P<0.05$, one-way ANOVA t-test) present were very different among the three groups while the N-glycoprotein types ($P>0.05$, one-way ANOVA t-test) were more similar. In particular, the difference between the 3D group and the tumor xenograft group is minimal (Fig 2C–2E). In general, the 3D-cultured breast cancer cells have shown different glycoproteomic characteristics compared to the 2D-cultured cells and xenografted tumors. In some aspects (such as the number of N-glycopeptides, N-glycans, N-glycoproteins, N-glycan compositions, and N-glycoprotein types), 3D-cultured breast cancer cells *in vitro* can better mimic xenografted tumors in mice *in vivo* compared to 2D-cultured breast cancer cells.

## Observed N-glycosylation differences among three groups

In order to illustrate the differences among the three groups, we further analyzed the characteristics of the intact N-glycopeptides, N-glycans, and N-glycoproteins. As shown in Fig 3A, the number of intact N-glycopeptides with 7~20 amino acid lengths (over 85%) in the 2D group was greater than that in the 3D group and the tumor xenograft group. Besides, the number of unique peptides in the 2D group was also larger (S2 Table). Interestingly, the number and the ratio of the N-glycan types (high-mannose, hybrid, and complex types) were very close among the three groups (Fig 3B). There were also some unique N-glycan compositions with putative structures for each group, and these unique N-glycan compositions (complex type) can be identified in two biological replicates. Specifically, the 2D group contained some unique sialylated high-antennary N-glycans (4/6) while the co-existence of N-acetylneuraminic acid (NeuAc) and N-glycolylneuraminic acid (NeuGc) in a single N-glycan ("mixed" high-antennary N-glycan) (5/6) was detected in the 3D group. In addition, unique polyfucosylated N-glycans were identified from the tumor xenograft groups (Fig 3B). For more complex models, the unique N-glycan compositions were simpler. We believe that these differences in N-glycan composition are due to differences in the cellular microenvironment.

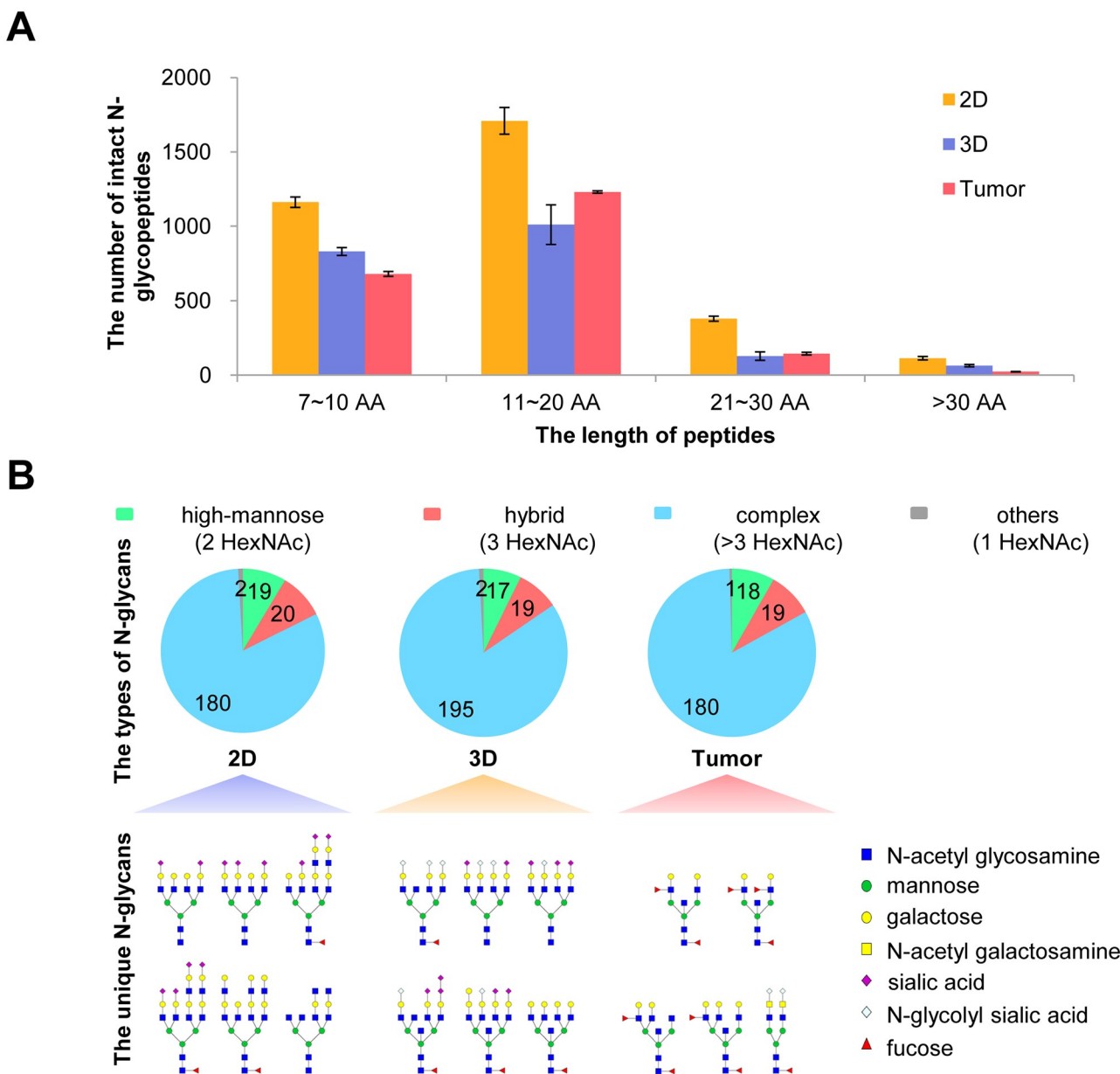

**Fig 3. Comparison of intact N-glycopeptides and N-glycans identified from 2D and 3D breast cancer cell cultures and xenografted tumors. A.** Number of intact N-glycopeptides with different lengths of amino acids (AA). **B.** Types of N-glycans and unique N-glycan compositions with putative structures identified from the 2D, 3D, and tumor xenograft groups.

At the N-glycoprotein level, 177 unique N-glycoproteins were identified from two biological replicates of the 2D group while 26 and 51 unique N-glycoproteins were identified from the 3D group and tumor xenograft groups, respectively (Fig 4A). These N-glycosylated proteins (including their intact N-glycopeptides) may serve as biomarkers for distinguishing cells grown in different types of cultures (2D, 3D, and tumor xenografts). These differentially expressed N-glycoproteins were then subjected to pathway enrichment analysis, and their statistical significance was determined by Fisher's exact test on the basis of the MSigDB pathway database [23]. As shown in Fig 4B, the pathway analysis revealed that the lysosome, cell surface

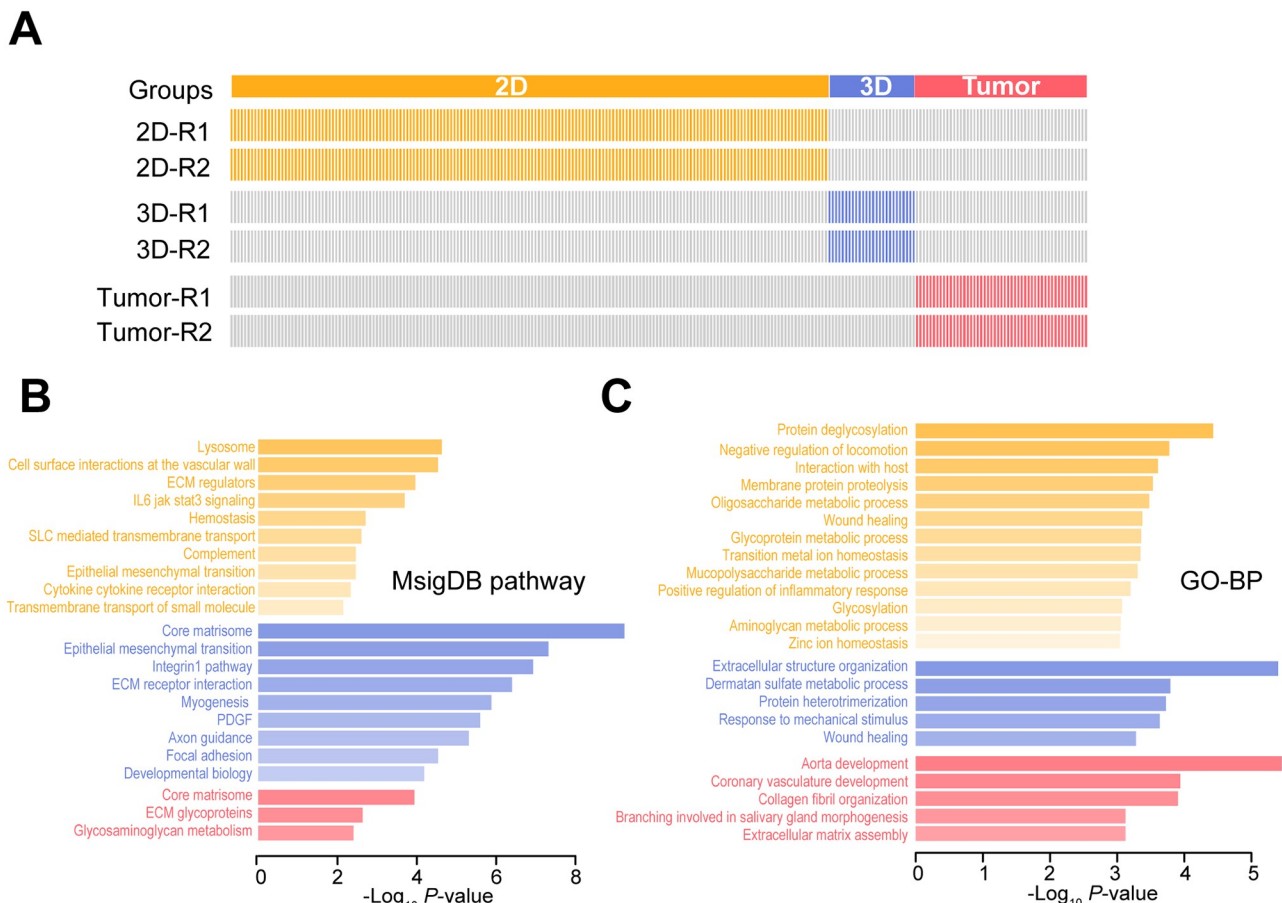

**Fig 4. Global analysis of differentially expressed N-glycoproteins identified from 2D and 3D breast cancer cell cultures and xenografted tumors.**
**A.** Heat map of unique N-glycoproteins expressed in two biological replications. Gray lines indicate the absence of the N-glycoprotein. Colored lines indicate the presence of the unique N-glycoprotein. **B.** Pathway analysis. **C.** Gene ontology biological process (GO-BP) enrichment analysis.

interactions at the vascular wall, ECM regulators, and other pathways were enriched in 2D-cultured cells while the core matrisome, epithelial mesenchymal transition, integrin-1 pathway, and other pathways were enriched in 3D-cultured cells. Furthermore, the core matrisome, ECM glycoproteins, and glycosaminoglycan metabolism pathways were enriched in the tumor xenograft group. These results suggest that glycosylation is very active in the lysosome and core matrisome pathways in different cell models. Gene ontology biological process (GO-BP) enrichment analysis of these identified unique N-glycoproteins was also performed to examine the relative biological processes where these N-glycoproteins are associated. As shown in Fig 4C, most unique N-glycoproteins were involved in protein deglycosylation (2D group), extracellular structure organization (3D group), and aorta development (tumor xenograft group). These results suggested that N-glycoproteins play an important role in cell culture, and that different cell growth microenvironments require different glycosylation expression patterns in N-glycoproteins.

## Conclusions

In this study, we presented an optimized workflow for in-depth N-glycosylation analysis by comparing different sources of trypsin, combinations of different proteases, and different

enrichment materials. Finally, we found that the combination of Trypsin (B)/Lys-C for digestion and Zic-HILIC for enrichment can significantly increase the number of identified intact N-glycopeptides and possesses better analytical reproducibility. Furthermore, the optimized workflow was applied initially to the intact N-glycopeptide analysis of 2D- and 3D-cultured breast cancer cells *in vitro* and xenografted tumors in mice. We identified 740 N-glycoproteins, 7229 N-glycopeptides, and 269 N-glycan compositions, where each of the three groups contained some unique intact N-glycopeptides, N-glycoproteins, and N-glycan compositions. Therefore, we propose that the differences in glycosylation patterns in proteins obtained from different culture models may affect *in vitro* studies tumor biology studies and drug evaluations using 2D- and 3D-cultured cells.

## Supporting information

**S1 Fig. The overlap of intact N-glycopeptides, N-glycoprotein, N-glycan compositions between two biological replicates of 2D (A) and 3D (B) cultured breast cancer cells in vitro and xenografted tumors (C) in mice in vivo.**
(DOCX)

**S1 Table. Intact N-glycopeptides of HepG2 identified from different groups.**
(XLSX)

**S2 Table. Intact N-glycopeptides identified from 2D and 3D breast cancer cell cultures and xenografted tumors.**
(XLSX)

## Author Contributions

**Data curation:** Yonghong Mao.

**Formal analysis:** Yonghong Mao.

**Funding acquisition:** Yong Zhang.

**Visualization:** Yang Zhao.

**Writing – original draft:** Yonghong Mao, Yong Zhang.

**Writing – review & editing:** Yong Zhang, Hao Yang.

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
