## [Decision Letter · Decision Letter 0]

25 Sep 2020

PONE-D-20-22961

In-depth Characterization and Comparison of the N-glycosylated Proteome of Two-dimensional- and Three-dimensional-cultured Breast Cancer Cells and Xenografted Tumors

PLOS ONE

Dear Dr. Zhang,

Thank you for submitting your manuscript to PLOS ONE. After careful consideration, we feel that it has merit but does not fully meet PLOS ONE’s publication criteria as it currently stands. Therefore, we invite you to submit a revised version of the manuscript that addresses the points raised during the review process.

Please see below for the comments from 2 reviewers.  As you will see, they raise some serious concerns about the current version of the manuscript.  Please note that novelty is not a criterion for publication in this journal, regardless of the reviewers' comments.  However, the clarity of presentation, full description of methods, citation of appropriate literature, and strength of experimental design are criteria for publication.  Please address those concerns listed below and resubmit your manuscript with the appropriate revisions.  Upon receipt, I will review the changes before sending the manuscript back out for review. Please let me know if you have any questions.

We look forward to receiving your revised manuscript.

Kind regards,

John Matthew Koomen, PhD

Academic Editor

PLOS ONE

Journal Requirements:

2. At this time, we request that you  please report additional details in your Methods section regarding animal care, as per our editorial guidelines:

(1) Please state the number of mice used in the study  

(2) Please provide details of animal welfare (e.g., shelter, food, water, environmental enrichment)

(3) Please describe the post-operative care received by the animals, including the frequency of monitoring and the criteria used to assess animal health and well-being.

Thank you for your attention to these requests.

3. Please provide additional information about each of the cell lines used in this work, including the source and any quality control testing procedures (authentication, characterisation, and mycoplasma testing). For more information, please see http://journals.plos.org/plosone/s/submission-guidelines#loc-cell-lines.

4. To comply with PLOS ONE submission guidelines, in your Methods section, please provide additional information regarding your statistical analyses. For more information on PLOS ONE's expectations for statistical reporting, please see https://journals.plos.org/plosone/s/submission-guidelines.#loc-statistical-reporting.

Reviewers' comments:

Reviewer's Responses to Questions

**Comments to the Author**

1. Is the manuscript technically sound, and do the data support the conclusions?

Reviewer #1: Yes

Reviewer #2: Partly

2. Has the statistical analysis been performed appropriately and rigorously? 

Reviewer #1: No

Reviewer #2: No

3. Have the authors made all data underlying the findings in their manuscript fully available?

Reviewer #1: Yes

Reviewer #2: Yes

4. Is the manuscript presented in an intelligible fashion and written in standard English?

Reviewer #1: Yes

Reviewer #2: Yes

5. Review Comments to the Author

Reviewer #1: The manuscript described here summarizes several of the previously published steps necessary to obtain a reasonable density of data from an LC-MS workflow which contain a reasonable recovery of peptides which contain N-glycan modifications. The data are fine, and well described and obtained. Unfortunately for this reviewer no new information is presented here, and a basic description of the enrichment is not made. The specific combination of results might be of more interest if the appropriate negative (no enrichment) controls were performed or the sample were deglycosylated with PNGase and then showing how much of this data are also lost (eg it's actually enriching)

The statistical analysis presented in figure 2b is not adequate for comparison of many means to many means. ANOVA or a welsh's or Dunnet/Tukey would be a more appropriate statistical treatment.

As such, I cannot recommend publication of this manuscript to to lack of novel findings, and lack of a novel method for the proposed analysis.

Reviewer #2: The manuscript is describing an enrichment approach to enhance LC-MS/MS analysis of N-glycopeptides derived from biological samples. The manuscript is also describing the use of analytical tools and mixed protease digestion approach to enhance the coverage of N-glycopeptides derived from biological samples. However, it has been demonstrated previously that HILIC [19-20] and a mix of proteases [5-8] are effective in providing adequate coverage of N-glycopeptides derived from biological samples. Therefore, the manuscript lacks novelty that warrants publication. The manuscript is not comparing the results to previously published work utilizing several enrichment approaches. The manuscript is not comparing their results to previously published work that employed multiple proteases. The biological study is interesting, and the data are intriguing. However, the statistical analysis is weakened by the use of only two biological replicates that were utilized in the study. The authors should conduct additional experiments to enhance the statistical analysis and focus the manuscript on the biological comparison. The method utilized in the study represents a combination of already published work by many researchers. The authors should do a better job referencing prior work conducted by many research groups. The quality of the figures is low. The authors should also expand the results and discussion section. Additional points that should be addressed by the authors are listed below.

Experimental Section

Intact N-glycopeptides enrichment, the protocol is using tryptic digest while the optimum method, as suggested in the manuscript, indicate the use of a mix of protease. Why were the enrichment methods not conducted using the mixed protease sample that, as suggested by the manuscript, provided the best approach to protease digest and analyze glycopeptides?

What is the source of the C-18 capillary column used?

Replace “The MS1 was analyzed with a...” with “The MS1 was performed employing…”

Replace “The MS2 was analyzed with an…” with “The MS2 was performed employing…”

Replace “…HCD collision type…” with “…HCD collision arbitrary value of 30%...”

What is meant by “stepped collision mode was turned on, with an energy difference of ±10%.”?

A ref. is missing in the following sentence “using Byonic software (version 3.6.0, Protein Metrics, Inc.) as described previously.”

The Orbitrap Fusion Lumos Mass Spectrometer’s mass accuracy is substantially better than 10 ppm. The authors should explain the rationale for using such mass accuracy.

Why did the authors only use two biological replicates? The statistical analysis is not valid when only two biological replicates are employed. You can always draw straight lines between two points.

6. PLOS authors have the option to publish the peer review history of their article (what does this mean?). If published, this will include your full peer review and any attached files.

Reviewer #1: No

Reviewer #2: No

---

## [Author Response · Author response to Decision Letter 0]

13 Oct 2020

Dear Professor John Matthew Koomen and Reviewers:

We are truly grateful for your encouraging feedback and insightful suggestions on our manuscript (PONE-D-20-22961). Base on these valuable comments, we have made careful modifications on the revised manuscript. Below you can find our detailed point-by-point responses to all the comments raised by the reviewers. The raw mass spectrometry (MS) data are publicly accessible at ProteomeXchange (ProteomeXchange.com) under Accession Number PXD020254 (Username: reviewer14082@ebi.ac.uk; Password: Ppbxc9FW). Changes made in the revised manuscript were highlighted by red colored text. Please do not hesitate to contact us if you need any further information. We look forward to hearing from you. 

Yours sincerely,

Hao Yang

Key Laboratory of Transplant Engineering and Immunology, West China Hospital, Sichuan University

Postal address: No. 1, Keyuan 4th Road, Gaopeng Avenue, Hi-tech Zone, Chengdu 610041, China

E-mail address: yanghao@scu.edu.cn

Journal Requirements:

Response: Thanks for your suggestion. We have modified this manuscript to meet PLOS ONE's style requirements. 

2. At this time, we request that you please report additional details in your Methods section regarding animal care, as per our editorial guidelines:

(1) Please state the number of mice used in the study 

(2) Please provide details of animal welfare (e.g., shelter, food, water, environmental enrichment)

(3) Please describe the post-operative care received by the animals, including the frequency of monitoring and the criteria used to assess animal health and well-being.

Thank you for your attention to these requests.

Response: We are grateful for your helpful suggestion. We have added the additional information regarding animal care to the revised manuscript. (Page 5-6)

3. Please provide additional information about each of the cell lines used in this work, including the source and any quality control testing procedures (authentication, characterisation, and mycoplasma testing). For more information, please see http://journals.plos.org/plosone/s/submission-guidelines#loc-cell-lines.

Response: Thank you for your suggestion. We have added the additional information about each of the cell lines used in this work to the revised manuscript. (Page 5-6)

4. To comply with PLOS ONE submission guidelines, in your Methods section, please provide additional information regarding your statistical analyses. For more information on PLOS ONE's expectations for statistical reporting, please see https://journals.plos.org/plosone/s/submission-guidelines.#loc-statistical-reporting.

Response: We have added the additional information regarding the statistical analyses to the revised manuscript. (Page 9)

Reviewer #1: 

The manuscript described here summarizes several of the previously published steps necessary to obtain a reasonable density of data from an LC-MS workflow which contain a reasonable recovery of peptides which contain N-glycan modifications. The data are fine, and well described and obtained. Unfortunately for this reviewer no new information is presented here, and a basic description of the enrichment is not made. The specific combination of results might be of more interest if the appropriate negative (no enrichment) controls were performed or the sample were deglycosylated with PNGase and then showing how much of this data are also lost (eg it's actually enriching). The statistical analysis presented in figure 2b is not adequate for comparison of many means to many means. ANOVA or a welsh's or Dunnet/Tukey would be a more appropriate statistical treatment. As such, I cannot recommend publication of this manuscript to to lack of novel findings, and lack of a novel method for the proposed analysis.

Response: We are truly grateful for the reviewer’s professional insights and valuable suggestions. Although we and other groups have done much work on glycoproteomics, there is a lack of systematic comparative analysis of these methods on glycoproteomics. In this paper, we focused on the optimized experimental method for intact N-glycopeptides identification from complex samples. As expected, the optimized sample preparation workflow (trypsin B/Lys-C (Enzyme & Spectrum,Beijing, China) digestion and Zic-HILIC (Fresh Bioscience, Shanghai, China) enrichment) prior to LC-MS/MS analysis, which can increase the number of identified intact N-glycopeptides and offer better analytical reproducibility (Figure 1). As suggested by the reviewer, we have added a detailed description of the enrichment in the revised manuscript. (Page 7)

 The specific combination of results (before enrichment, after enrichment and deglycosylated with PNGase) have been reported in our previous work (J. Proteome Res. 2020, 19, 2539−2552; J. Proteome Res. 2020, 19, 655−666). Before enrichment, there are less 10% intact N-glycopeptides can be identified from complex samples. After enrichment, the ratio of intact N-glycopeptides will increase to over 80%.

 In this paper, six replicates (two biological replicates and three technical replicates) for each group (2D, 3D or Tumor group) were employed. We used Student's t-test for statistical comparison between two groups. As suggested by the reviewer, we have added ANOVA test for statistical comparison among three groups. (Page 12)

Reviewer #2: 

1-The manuscript is describing an enrichment approach to enhance LC-MS/MS analysis of N-glycopeptides derived from biological samples. The manuscript is also describing the use of analytical tools and mixed protease digestion approach to enhance the coverage of N-glycopeptides derived from biological samples. However, it has been demonstrated previously that HILIC [19-20] and a mix of proteases [5-8] are effective in providing adequate coverage of N-glycopeptides derived from biological samples. Therefore, the manuscript lacks novelty that warrants publication. The manuscript is not comparing the results to previously published work utilizing several enrichment approaches. The manuscript is not comparing their results to previously published work that employed multiple proteases. The biological study is interesting, and the data are intriguing. However, the statistical analysis is weakened by the use of only two biological replicates that were utilized in the study. The authors should conduct additional experiments to enhance the statistical analysis and focus the manuscript on the biological comparison. The method utilized in the study represents a combination of already published work by many researchers. The authors should do a better job referencing prior work conducted by many research groups. The quality of the figures is low. The authors should also expand the results and discussion section. Additional points that should be addressed by the authors are listed below.

Response: We appreciate the reviewer’s professional comments and valuable suggestions. The characterization of protein glycosylation is essential and challenging. Due to the lack of suitable analytical methods for in-depth analyses of intact N-glycopeptides, it is hard to understand how glycosylation contributes biologically to human. In this paper, we presented an optimized sample preparation workflow (trypsin B/Lys-C (Enzyme & Spectrum,Beijing, China) digestion and Zic-HILIC (Fresh Bioscience, Shanghai, China) enrichment) prior to LC-MS/MS analysis, which can increase the number of identified intact N-glycopeptides and offer better analytical reproducibility. 

 Although HILIC[19-20] and a mix of proteases [5-8] also are effective, those methods are different from ours. We compared the most commonly used method (trypsin A-HILIC) and other approaches (trypsin A-Zic-HILIC, trypsin B-HILIC, trypsin B-Zic-HILIC, trypsin B/Lys-C-HILIC, trypsin B/Lys-C-Zic-HILIC). Our results revealed that the combined method, trypsin B/Lys-C-Zic-HILIC, can significantly increase the number of identified intact N-glycopeptides and offer better analytical reproducibility (Figure 1). 

 Then, the optimized workflow was applied to analyze native intact N-glycopeptides from 2D and 3D cultured breast cancer cells in vitro and xenografted tumors in mice. The number of intact N-glycopeptides, N-glycan compositions, and N-glycoproteins identified from two biological replicates and three technical replicates for each sample (2D and 3D-cultured cells and tumor tissues). There were 18 RAW data produced by LC-MS/MS in total. We used Student's t-test for statistical comparison between two groups and employed ANOVA test for statistical comparison among three groups (Page 12). 

 As suggested by the reviewer, we have provided high quality figures and expanded the results and discussion section (Page 9-14).

2-Experimental Section

Intact N-glycopeptides enrichment, the protocol is using tryptic digest while the optimum method, as suggested in the manuscript, indicate the use of a mix of protease. Why were the enrichment methods not conducted using the mixed protease sample that, as suggested by the manuscript, provided the best approach to protease digest and analyze glycopeptides?

Response: We thank the reviewer's careful review. For the digestion of complex biological samples, the optimal method is trypsin (B)/Lys-C digestion combined with Zic-HILIC enrichment. We have modified the unclear description in the revised manuscript. (Page 7)

3-What is the source of the C-18 capillary column used?

Response: Thanks for the reviewer's careful review. The 75-μm-inner-diameter C-18 capillary column with a length of 20 cm (ReproSil-Pur C18-AQ, 1.9 μm; Dr Maisch) was prepared by ourselves. (Page 5)

4-Replace “The MS1 was analyzed with a...” with “The MS1 was performed employing…”;Replace “The MS2 was analyzed with an…” with “The MS2 was performed employing…”;Replace “…HCD collision type…” with “…HCD collision arbitrary value of 30%...”.

Response: Thank you very much for your careful review and valuable suggestions. We have modified these sentences in the revised manuscript. (Page 7-8)

5-What is meant by “stepped collision mode was turned on, with an energy difference of ±10%.”?

Response: Thanks for your question. We have modified this sentence "The stepped collision mode was turned on with an energy difference of ±10% (20-30-40%)" in the revised manuscript. (Page 8)

6-A ref. is missing in the following sentence “using Byonic software (version 3.6.0, Protein Metrics, Inc.) as described previously.”

Response: We appreciate the reviewer for the careful review. We have added the reference in the revised manuscript. (Page 8)

7-The Orbitrap Fusion Lumos Mass Spectrometer’s mass accuracy is substantially better than 10 ppm. The authors should explain the rationale for using such mass accuracy.

Response: We are truly grateful for the reviewer’s professional insights. We generally set precursor mass tolerance to 10 ppm, and fragment mass tolerance to 20 ppm for MS2 acquired by Orbitrap according to the manufacturer’s suggestion (Byonic™: Mass Tolerance Settings, www.proteinmetrics.com). The data quality was controlled by 1% FDR at the proteins level, and score cutoff of 200 for glycopeptides. In fact, most of the identified spectra are distributed between -5 ppm and +5 ppm (Table S1 and S2).

8-Why did the authors only use two biological replicates? The statistical analysis is not valid when only two biological replicates are employed. You can always draw straight lines between two points.

Response: Thanks for the reviewer's question and comments. We agree with the reviewer’s point of view. More biological replicates will provide more accurate differential glycopeptides among different groups, which will be helpful for understanding the contributions of glycoproteins in different cell growth conditions. In this paper, we predominately focused an optimized experimental method for intact N-glycopeptides identification from complex samples. Considering the animal welfare, we used two biological replicates for each group. However, we also performed three technical replicates for each group so that we could perform the statistical analysis. We appreciate the reviewer’s suggestion and we will use more biological replicates to validate the differential glycoproteins and explore their functions in future studies.

---

## [Decision Letter · Decision Letter 1]

26 Nov 2020

In-depth Characterization and Comparison of the N-glycosylated Proteome of Two-dimensional- and Three-dimensional-cultured Breast Cancer Cells and Xenografted Tumors

PONE-D-20-22961R1

Dear Dr. Zhang,

We’re pleased to inform you that your manuscript has been judged scientifically suitable for publication and will be formally accepted for publication once it meets all outstanding technical requirements.

Kind regards,

John Matthew Koomen, PhD

Academic Editor

PLOS ONE

Additional Editor Comments (optional):

Thank you for revising your previous manuscript. Please see the note from the reviewer about the p values in the figures. Is it optional for you to change the format, but please make sure that the figures are clear and concise.

Reviewers' comments:

Reviewer's Responses to Questions

**Comments to the Author**

1. If the authors have adequately addressed your comments raised in a previous round of review and you feel that this manuscript is now acceptable for publication, you may indicate that here to bypass the “Comments to the Author” section, enter your conflict of interest statement in the “Confidential to Editor” section, and submit your "Accept" recommendation.

Reviewer #1: All comments have been addressed

2. Is the manuscript technically sound, and do the data support the conclusions?

Reviewer #1: Yes

3. Has the statistical analysis been performed appropriately and rigorously? 

Reviewer #1: Yes

4. Have the authors made all data underlying the findings in their manuscript fully available?

Reviewer #1: Yes

5. Is the manuscript presented in an intelligible fashion and written in standard English?

Reviewer #1: Yes

6. Review Comments to the Author

Reviewer #1: This manuscript is substantially improved. I firmly believe the novel findings here are limited; but this is not the driving threshold for this journal.

Figure 2B. The P-values in the graph are very busy. Change them to significant * ** or not and put the numerical values in the figure legend to clean it up.

7. PLOS authors have the option to publish the peer review history of their article (what does this mean?). If published, this will include your full peer review and any attached files.

Reviewer #1: No

---

## [Editor Report · Acceptance letter]

2 Dec 2020

PONE-D-20-22961R1 

In-depth Characterization and Comparison of the N-glycosylated Proteome of Two-dimensional- and Three-dimensional-cultured Breast Cancer Cells and Xenografted Tumors 

Dear Dr. Zhang:

I'm pleased to inform you that your manuscript has been deemed suitable for publication in PLOS ONE. Congratulations! Your manuscript is now with our production department. 

Kind regards, 

on behalf of

Dr. John Matthew Koomen 

Academic Editor

PLOS ONE